# Understanding Open Defecation in the Age of *Swachh Bharat Abhiyan*: Agency, Accountability, and Anger in Rural Bihar

**DOI:** 10.3390/ijerph17041384

**Published:** 2020-02-21

**Authors:** Anoop Jain, Ashley Wagner, Claire Snell-Rood, Isha Ray

**Affiliations:** 1Civil & Environmental Engineering, Stanford University, Palo Alto, CA 94305, USA; 2School of Public Health, U.C. Berkeley, Berkeley, CA 74707, USA; ashley_wagner@berkeley.edu (A.W.); clairesnellrood@berkeley.edu (C.S.-R.); 3Department of City and Regional Planning, U.C. Berkeley, Berkeley, CA 94704, USA; 4Energy & Resources Group, U.C. Berkeley, Berkeley, CA 94704, USA; isharay@berkeley.edu

**Keywords:** open defecation, sanitation, social determinants, Swachh Bharat Abhiyan, India, environmental health, Sustainable Development Goals

## Abstract

Swachh Bharat Abhiyan, India’s flagship sanitation intervention, set out to end open defecation by October 2019. While the program improved toilet coverage nationally, large regional disparities in construction and use remain. Our study used ethnographic methods to explore perspectives on open defecation and latrine use, and the socio-economic and political reasons for these perspectives, in rural Bihar. We draw on insights from social epidemiology and political ecology to explore the structural determinants of latrine ownership and use. Though researchers have often pointed to rural residents’ preference for open defecation, we found that people were aware of its many risks. We also found that (i) while sanitation research and “behavior change” campaigns often conflate the reluctance to adopt latrines with a preference for open defecation, this is an erroneous conflation; (ii) a subsidy can help (some) households to construct latrines but the amount of the subsidy and the manner of its disbursement are key to its usefulness; and (iii) widespread resentment towards what many rural residents view as a development bias against rural areas reinforces distrust towards the government overall and its Swachh Bharat Abhiyan-funded latrines in particular. These social-structural explanations for the slow uptake of sanitation in rural Bihar (and potentially elsewhere) deserve more attention in sanitation research and promotion efforts.

## 1. Introduction

In 2015, the United Nations General Assembly announced 17 Sustainable Development Goals (SDGs) to be met by 2030. One of these goals, SDG 6, calls on the global community to “ensure availability and sustainable management of water and sanitation for all” [1]. A particular concern was the persistence of open defecation (OD) in the Global South. Open defecation is defined as the disposal of human feces in open areas, such as fields, forests, road side, beaches, and open bodies of water [2]. 

OD results in the spread of untreated fecal matter throughout the environment and is associated with a variety of negative health outcomes. Fecal contamination is associated with diarrheal diseases, trachoma and schistosomiasis [3,4]. Exposure to fecal bacteria has been linked to stunting, a measure of linear growth retardation that is often used as a predictor of long-term educational and economic outcomes [5]. In addition, OD, and inadequate sanitation more generally, is associated with psychosocial stress [6]. There is thus widespread consensus among health researchers that adequate sanitation and hygiene are key determinants of health [7,8].

In India, the World Health Organization’s (WHO) Joint Monitoring Programme estimated that ~520 million people were regularly defecating in the open in 2015 [2]. The problem is of particular concern in rural areas, where 69% of households reported that they did not own a latrine in 2011 [9]. Unimproved sanitation accounts for an estimated 2.4% of the total burden of disease, expressed as Disability-Adjusted Life Years [10]. India is also home to 28.5% of the world’s stunted population; studies suggest that districts with a higher prevalence of OD are more likely to have more stunted children [11]. Furthermore, inadequate access to sanitation leads to psychosocial stress among women and girls due to environmental barriers, such as physical distance and unsafe latrines, social factors, and fears of sexual violence [12]. UNICEF estimates that inadequate sanitation costs India $189 billion annually, or 7.9% of GDP, as a result of medical costs, lost tourism, and losses in productivity [13]. 

Two primary approaches have been developed to address OD; most large-scale sanitation promotion programs use a mix of both. Demand-side interventions aim to generate demand for latrines through information campaigns and behavior change programs, while supply-side interventions prioritize providing sanitation infrastructure [14,15]. Researchers have argued that building or subsidizing toilet construction is ineffective and unsustainable without behavior change [16,17]; that many Indians have a preference for OD over using latrines [18]; and that more efforts, therefore, are needed towards education, awareness-building and peer pressure to build demand [18,19,20]. The current literature on sanitation uptake therefore emphasizes demand-side approaches, and thus individual- and community-level attitudes and preferences as determinants of latrine adoption (see Dreibelbis et al., 2015) [21]. Researchers argue that positive social marketing messages are more powerful in encouraging behavior change than negative disease-related messages [7,22]. However, the information and education components of India’s sanitation promotion efforts continue to emphasize health information towards sanitation uptake [23]. Studies have also argued that people in rural India prefer OD and do not want to use a latrine even if they have one [18,24,25]. The reasons given for this preference range from the difficulty of access to the belief that going in the open is healthier than going in a confined space. Recent work also suggests that, in rural India, the difficulty of emptying full pits contributes to a reluctance to use latrines [26,27]. The emphasis on household-level choices has resulted in several interventions focused on motivating individuals and communities to change their practices through education, incentives and peer pressure. Social and structural drivers of latrine use or disuse, until recently, remained under-emphasized [28]. 

The Government of India has been trying to eliminate open defecation since the 1980s. Its latest sanitation scheme is a program called Swachh Bharat Abhiyan, which aimed to achieve universal sanitation coverage and use by 2 October 2019 (2 October 2019 was the 150th anniversary of Mahatma Gandhi’s birth and this date was chosen as a tribute to him and his dream of promoting cleanliness and hygiene throughout India). The rural division of the program—Swachh Bharat Abhiyan-Gramin (SBA-G)—focuses on behavior change communication, such as advertisements on TV, radio, and social media featuring local champions and celebrities, which are intended to galvanize communities to adopt safe sanitation practices. Rural households living below the poverty line, and certain groups such as people with disabilities and widows, that construct a latrine are also eligible for an incentive of up to 12,000 Indian Rupees (approximately $174 in 2019 United States Dollars). Households can choose what type of latrine they build, but the SBA-G guidelines advocate for a twin-pit pour flush latrine, saying that they are low cost, safe to empty, and consume little water [23]. This approach—using behavior change theory to influence individual, household, or community-level practices and promoting low-cost sanitation technologies—is the norm in many settings around the world [28].

Government reports claim that 96 million latrines have been constructed under SBA-G and that every state now has 100% sanitation coverage [29], but researchers both before and after SBA-G have expressed skepticism about how open defecation free (ODF) status is verified and about the statistics the government publishes [27,30]. One study found that only one in eight of the villages declared to be ODF in 2018 actually had universal sanitation coverage [31]. Additionally, while OD has declined significantly in rural areas and there have been major national improvements in latrine coverage, large interstate variations in uptake remain. For example, sanitary latrine coverage in rural Bihar was at 48% compared to a reported 100% coverage in rural Kerala in 2017 [32]. It is clear that, despite the national attention to SBA-G and the progress it has made, there are gaps in coverage and use across regions. In this paper we ask: what social and political factors can help to understand the persistence of OD in rural Bihar, despite many years of latrine promotion under SBA-G? 

### Under the Lid of Swachh Bharat Abhiyan

This paper builds on research that has explored the socio-economic context around OD and the politics of access to sanitation infrastructure. Since focusing on individual behavior change has been only partially effective at eliminating OD [33,34], we examine social-structural constraints to latrine use to better understand the broader context of rural India’s sanitation challenge. We draw on insights from social epidemiology, with its explicit focus on health disparities and equity [15,35], and political ecology [36,37], which highlights the cultural and political determinants of access to services. Both of these approaches urge researchers to look beyond individual behaviors to the structural factors—such as economic status and social identities—that shape how people make health-related choices.

We present an exploratory study of perspectives on open defecation and latrine use, and the socio-economic and political reasons for these perspectives, in rural Bihar, one of the worst-performing states under SBA-G. Our primary methods were fieldwork and interviews with residents in the Supaul District, in which we asked about their experiences with trying to build and use a latrine. We found that our participants were well aware of the many risks of OD. Our informal discussions and formal interviews revealed that (i) while sanitation research and “behavior change” campaigns often conflate the reluctance to adopt latrines with a preference for OD, this is an erroneous conflation; (ii) a subsidy can help (some) households to construct latrines but the amount of the subsidy and the manner of its disbursement are key to its usefulness; and (iii) widespread resentment towards what many rural residents view as a development bias against “small” rural people may reinforce distrust towards the government overall and its SBA-funded latrines in particular. These social-political explanations for the slow uptake of sanitation in rural Bihar (and potentially elsewhere) deserve more attention in sanitation research and promotion efforts.

India has been considered a “problem” for latrine use from colonial times: blunt declarations by foreign health workers such as “the Hindoo cannot be made to use a latrine” (see Cosgrove, 1909, p.67) [38] have since been replaced by “fixing dreadful sanitation in India requires not just building lavatories but changing habits” (*The Economist* in 2014) [39]. SBA-G built on many past efforts to eliminate OD in India, which largely failed to meet their targets [14,30]. 

Since 1999, India has officially favored a demand-side approach, first via the Government-initiated Total Sanitation Campaign (TSC). The TSC promoted health education combined with hardware subsidies to generate demand for latrines. Critiques of the TSC have since shown that the program remained top–down, and that it was unsuccessful in promoting latrine construction and use [40,41]. In 2012, this campaign was renamed Nirmal Bharat Abhiyan and a Community-Led Total Sanitation (CLTS) component, which uses leadership training and peer-pressure against OD to trigger behavior change, was added [36]. However, the subsidies for low-income households, which CLTS programs typically eschew, were retained.

In 2014, the Government of India revamped Nirmal Bharat Abhiyan as SBA, financing it generously, and significantly ramping up its social marketing. SBA also stepped up the pace of public, community, and school latrine construction, though its primary focus remained on individual household latrines [23]. SBA-G also deployed Swachhagrahis to teach community members about the consequences of OD and about good hygiene practices. The SBA-G guidelines state that “It is important that ODF [open defecation free status] has been achieved through a focus on collective behavior change and demand generation, and not through supply-driven mode,” [23] (p. 11) and that “sanitation is primarily a behavioral and demand-driven issue…” [23] (p. 16). 

SBA-G gives states the flexibility to determine their own implementation policies, including how latrines should be constructed and how incentives are distributed. States have the option to provide the household incentive either at the pre-construction stage or upon completion of construction. States may also choose to give the incentives to households or to the community as a whole once it has achieved ODF status. If the community incentive model is selected, the incentive “may only be released after the village unit is open defecation free for a significant length of time” [23] (p. 18). This practice of deliberately delaying the release of the reimbursement to ensure that people will not revert to defecating in the open is reiterated as a sustainability strategy throughout the program implementation guide [23].

Most existing behavior change models take a similar individual-level, demand-side approach and tend to underplay the multi-level interactions that shape whether or not someone will adopt a new behavior and sustain it [28]. While this approach has had some success in reducing OD in the short term, there is little evidence that it leads to sustained improvement [33,34]. This pushes us to look to the broader structural (i.e., political and social) determinants that are equally essential for latrine adoption [28,42]. Doing so can generate new insights into the associations between social determinants and disparities in latrine ownership and use in India, and thus generate critical knowledge on how to design better sanitation interventions.

## 2. Methods 

Our study state, Bihar, had a population of over 104 million people in 2011 [9]. It is one of the fastest growing economies among the low-income states in India, but continues to lag behind India’s average in terms of social, economic, and health outcomes [43]. Approximately 51% of the state’s Scheduled Caste (SC) households—the lowest rung of the traditional caste ladder—live in poverty. In 2016, approximately 31% of households had access to improved sanitation and 68% of households defecated in the open [43]. OD is highest among households in the lowest two income quintiles (81% and 82%, respectively), but is also high in the highest income quintile (49%) [43].

Our research was conducted in the Supaul District of Bihar, where 95% of the 2.2 million residents live in villages [9]. In 2015–2016, only 15.3% of rural households in Supaul had access to improved sanitation [44]. Within Supaul, our study participants were from three villages: Sukhpur, Karanpur, and Baukar. 

Our primary data collection method was a series of exploratory, open-ended discussions, in groups and one-on-one, on sanitation practices, attitudes towards OD, latrine ownership and use, and the reasons for non-adoption and/or non-use despite the financial assistance and social marketing efforts of SBA-G. Our research team asked participants whether or not their household owned a latrine; if no, we asked why not, whether they knew about the benefits of latrine ownership, use, and waste management, and what the government could do to help more people gain access to household latrines. If the respondent did have a latrine, the research team asked when it was built, whether it was used consistently, what it took to manage the waste, and why they thought others might not own a latrine. 

We ensured variation in age (but excluded minors under the age of 18), gender, caste, household latrine ownership, occupation, and education among our participants, as each of these characteristics could be associated with latrine ownership and OD preferences (see Table 1). Based on these discussions, we followed up with extended, semi-structured interviews with seven men and six women; these interviewees, in effect, acted as key informants for our study. We also conducted two focus groups, exclusively with men, because men made most of the financial decisions (such as whether or not to construct a latrine) within all of the households we encountered. Each focus group included four men. While there was demographic variation between participants, we maintained homogeneity with regards to caste, gender, and occupation within focus groups. This was done to ensure that participants felt comfortable sharing their views in a group setting without fear of prejudice or shame. 

### Data Collection and Analysis

Data collection took place between June and August of 2018. All the interviews and focus groups were conducted in Hindi or Maithili (the language spoken in north Bihar) by the two lead researchers, with the help of a translator proficient in English, Hindi, and Maithili. Researchers recruited participants through snowball sampling, as our intention in this study was to focus on the similarities in their responses [45]. Overall, interviews lasted an average of 17 min, running shorter or longer depending on the participant’s level of interest. The two focus groups with men were longer, each lasting almost an hour. Each interview was digitally recorded, and then transcribed from Hindi/Maithili to English. The translator worked alongside the researchers, listening to the audio recordings and looking over the transcripts, to ensure that each interview was being properly translated and transcribed. A randomly chosen set of recordings and transcriptions was given to a second translator as a quality control step. All participants gave informed (verbal) consent to participating in our study, and our study design was approved by UC Berkeley’s Office for the Protection of Human Subjects (Protocol ID 2016-08-9092).

We used a thematic analysis approach, systematically identifying and organizing themes that emerged, to analyze our data [46]. This process was conducted over multiple stages. Initially, we reviewed a subset of data and began developing our codebook. Each researcher wrote notes and memos, and we collaboratively discussed the patterns that we noticed around discourse surrounding sanitation, participants’ financial realities creating barriers to toilet construction, lived sanitation experiences, and any expressions of distrust of government. This formed the foundation of the first draft of our codebook. After all of our interviews were completed and transcribed, we reviewed them and refined our codebook and grouped our codes into broader thematic categories. Two researchers coded each transcript line-by-line using the qualitative data analysis software ATLAS.ti (version 8.3.1, www.atlasti.com). The research team came together after this to ensure intercoder reliability. This enabled us to further refine the codebook and our coding scheme (when and where we applied codes). Each researcher then used the revised codebook to re-code each transcript, and the connections that arose informed our major findings. 

## 3. Results

We present three key results in this paper. Our first result challenges the widespread notion that low rates of latrine ownership and use are a function of a preference for open defecation; this may be the case for some regions, but low latrine adoption cannot be conflated with a preference for OD. Rather, as our second and third results show, people are often unable to own a latrine despite SBA-G because (i) the design of SBA-G’s subsidy scheme, along with the inadequate subsidy amount, is impractical for India’s rural poor; and (ii) there is a perceived anti-poor and anti-rural bias in public service provision, which contributes to the distrust of government and government sanitation programs in rural regions, and that generally discourages participation.

### 3.1. Open Defecation Is A Necessity and Not A “Preference”

Our participants uniformly denied that low rates of latrine ownership and use could be explained by a preference for OD. They described open defecation as “majburi”—a Hindi word meaning a coerced action, or something that must be endured. Our participants talked explicitly about feeling forced to defecate in the open: “Going outside does not feel good, but what will we do?” A man who used to defecate in the open simply said, “We used to feel a lot of shame.” Another participant, whose family now owns and regularly uses a latrine, agreed: “Going in the open was majburi for us. It was very bad.” Younger participants talked about how wealthier, high-caste community members were able to build and use latrines, while those who were poor and low-caste were “forced to endure” life without a latrine.

Some researchers have reported that people in India might prefer defecating in the open because it could promote good health by creating an opportunity to go for walks in the fresh air, and/or allowing people to maintain “pure” households free from fecal contamination, which is considered taboo in Hindu texts [47]. We did not encounter these attitudes in our study sites. Our participants refuted the first notion vehemently: “No, no, people do not defecate in the open because they want fresh air, they go because it is a necessity.” Others pointed out that people living in households with latrines are “cooking, sleeping, and living fine,” and are not worried about polluting their homes so long as the latrine is “kept clean and covered.”

Our participants also demonstrated a keen awareness about the benefits of latrine use and the dangers associated with OD; they suggested that more information and education were not needed, and that lack of awareness was not a reason for open defecation to be “preferred” to latrine use. Interviewees explained how latrines can help ensure a healthy, hygienic life, and they had a clear understanding of how they prevent disease transmission through flies. One participant commented: “We all think that there should be latrines so that we are protected from disease.” However, if the only choice was pits near the home that might not safely contain feces—an option perceived to be temporary and inadequate—then “going in the open is a better option.” The men we interviewed also admitted to a sense of shame that so many village women were forced to cover their faces to protect their privacy and dignity when they went on the side of the road every morning: “This is wrong, sir. Solid wrong. This should not be happening.”

### 3.2. Subsidy Model for Latrine Construction is Inadequate

Our interviews corroborated other research that has shown the subsidy for latrine construction provided through SBA-G (12,000 Indian Rupees) to be inadequate to cover the costs for poor households [30]. The government recommends twin-pit latrines for rural households, which families have to empty once they are full. As a result, many people want larger pits as they will not fill up quickly. According to our participants, however, building larger pits increases the cost of construction beyond the subsidy amount. For example: “You just imagine the government is giving 12,000—if I am a poor man and I don’t want to invest too much, I will make a small pit. And a big man will spend 50,000 and build a big pit. The government should think for itself, how big a latrine can someone possibly build with 12,000?” Most interviewees said that building a “proper latrine” (i.e., with an adequate pit size) was well outside their financial means. Emptying the waste through hired labor or tankers costs money: “[A tanker] is a big cost. They take 2000 rupees, 2500 rupees…if poor people don’t have it, of course they’ll defecate outside.” In some cases, participants said that they might consider emptying the waste themselves to save money, but that this also had its challenges. Where, for instance, was a large volume of waste to be put? 

Poor people, our interviewees said, are forced to spend whatever little they have on necessities such as food, leaving nothing for latrines. In other cases, the care of a sick family member used up whatever spare money there was, or illness left families “buried in debt.” Several women we spoke to want a latrine but they were not the ones making the financial decisions for their households. For example, one young woman said that her family saved any spare money for her marriage, since women’s families are still expected to pay for marriage and its accompanying dowry in North India. These are the realities that our participants felt that the government did not fully understand, because officials do not “come house to house to see the situation,” but pressure families to invest in latrines regardless. One participant argued: “The government says, ‘live in cleanliness, drink good water, don’t defecate in the open, do all these important things’…But if you have no money, then what?”

We found that, even for those households that were willing to cover the costs beyond the subsidy, SBA-G’s incentive-delivery method was not practical. In Bihar, to receive the subsidy, people had to first pay for latrine construction out of pocket, and then submit a picture and forms to prove that they had done so. Almost everyone we interviewed raised questions about poor people’s ability to pay for these costs upfront: “How will people who don’t have money first build a latrine?” Many participants expressed fears that they would not receive the full subsidy because of corruption. For example, some participants had heard that they would have to pay 16% of the reimbursement to the local officials just to get their SBA-G paperwork processed. In addition, participants reported that the rules regarding when and how the reimbursement would be issued kept changing. They did not know how long it would take local authorities to transfer the subsidy to their bank accounts: “If the money doesn’t come [from the government] for five years, what’s the benefit then?” One participant said that “the government” told him that “until 95% of the ward has latrines, none of the households will be eligible for the reimbursement.” He was nervous to pay money upfront to build a latrine because he was unsure when, or if, his neighbors would also build latrines. Given the SBA-G guidelines, states and districts do have the flexibility to delay the disbursement of reimbursements until a certain percentage of households in a village have constructed a latrine [23]. 

Participants commented that using a loan to pay for reimbursable latrine construction costs was also untenable. One participant articulated this fear, saying, “some poor family might take a loan that comes with interest. They can spend that money to build a latrine and fill out the required paperwork and documents to get the reimbursement, but they have no idea when the reimbursement will come. It could take one, two, or even three years, so what about all the interest?” Yet others had little faith that the reimbursement would ever come; one man simply said, “there is no guarantee.” In sum, the subsidy amount, financial realities of the rural poor, and the mechanism through which the subsidy is delivered all combined to make SBA-G’s incentive model impractical for universal coverage. 

### 3.3. A Perceived Anti-Poor and Anti-Rural Bias Hinders Latrine Uptake

A significant emerging theme in our results was participants’ description of an overall distrust of government that shaped not only their participation in SBA-G, but that was foundational toward understanding themselves as disenfranchised citizens. Interviews revealed a widespread perception that SBA-G, allegedly like other development initiatives, prioritized development in urban centers and was less willing to help India’s rural citizens. Participants’ generalized anger and resentment towards the government critically influenced their willingness and ability to trust its programs such as SBA-G, or to construct a latrine through SBA-G. 

For example, though much of urban India remains unsewered [48], there are significant differences in the waste management technologies available in urban and rural areas. In Bihar, 61% of rural households do not have access to a wastewater system, compared to 29% of households in urban areas [49]. Cities and towns have better access to mechanized cleaning of septic tanks and sewers, via private or municipal services, than rural areas do. These and other service disparities were visible to those who had been to cities (or who had family members living in cities), and they angered many participants. Some brought up their experiences in cities such as Delhi, where they saw “pipes, drains, and sewers.” Such experiences fueled resentment towards urban areas and prompted our participants to question why they would have to settle for second-rate infrastructure. One interviewee said that he had emptied the waste from his pit, but had had no option but to put it on another part of his land (a reality for many families in the area); he became afraid that this waste could spread to his home. More than one participant expressed disgust at the idea of cleaning out their own latrine pits, saying they would “probably vomit if [they] had to do it.”

Our focus groups came to a consensus that the lack of waste management was perhaps the greatest barrier to sanitation in rural Bihar (corroborating Coffey and Spears, 2017) and that a disposal system in rural areas would benefit everyone. Ideally, this would be a sewer, but anything hygienic that would not need too much “management” on their part would work. They thought that rural households might “no longer have to empty their pits out” and that people would “feel some relief” if they did not have to manage their own waste. Our participants pointed out that people in Delhi and Mumbai had pipes and were not expected to manage their own waste. If they could not get a sewer system, a village-level tanker would do: “If there was a tanker for the village…they would go door-to-door to empty.” Furthermore, “whether you are rich or poor, this will help you.” Our research team asked the focus groups why they thought that the government had not brought in vacuum-pump operated tankers for cleaning out pits when they all agreed that such a system would help to achieve a “Clean India.” The response was twofold: “They do not care about these things,” and “because India is clean for big people, it isn’t clean for little people.” Alluding to a more general sense of distrust, one man said that his community had “no faith” in the government. Another participant questioned why people continued to support elected officials, saying, “The government isn’t doing proper development work, so why do we help them win? They don’t give us anything.”

## 4. Discussion

Our goal was to examine the broader social and political factors influencing latrine adoption that have received less attention than individual-level determinants of sanitation outcomes, such as knowledge, preferences and habits endogenous to individuals. Currently, these factors undergird community-focused interventions such as SBA-G and CLTS. Individually-oriented interventions focused on behavior change leave households accountable for the failure to adopt safe sanitation practices. Education and social marketing have been only modestly successful in the sustained reduction of OD in India or elsewhere [34,50]. Behavior change interventions have also been critiqued because, for those who cannot change their behavior due to structural reasons, they generate “additional social stigma for failing to maintain new social norms regarding health behaviors” [36] (p. 194). At their worst, these behavior change interventions have been coercive, and cavalier with respect to individuals’ rights and privacy [7,30,51]. For all these reasons, research as well as policy needs to take seriously the social-political determinants of latrine adoption if OD is to be eliminated.

We oriented our field research towards elucidating the social determinants of latrine ownership and use in a setting with challenges. Social determinants are the economic and social systems that are shaped by and implemented through government interactions and international agreements [52,53], all of which are “upstream” of an individual’s attitudes and preferences. Political and environmental determinants include access to water and land, and government or non-governmental organization policies [37,54]. 

Our participants alluded to upstream determinants of latrine ownership in several ways. They argued that non-adoption of latrines was not due to preferences but competing financial demands [14,15]; gender inequities that prioritized the wishes of the male head of household [55,56,57]; and the uncertainty of a post-construction reimbursement. Our work does not contradict other research that has found a preference for OD in rural India [16,18,26]. Rather, the frequent reference to “majburi” indicates that OD was more a necessity than a preferred option in Supaul, which undermines the common policy conflation of the observed persistence of OD with its alleged preference. Additionally, our participants pointed to the shortcomings of SBA-G’s payment mechanism—with the subsidy following construction—which did not incentivize latrine construction, saying that the government is too slow in processing reimbursements. In effect, our participants were saying that the specific design of the policy does not account for their poverty, and that if the policy were to be redesigned, at least some households would build a latrine despite financial stress. This is important given that various studies confirm that well-designed subsidies can play an important role in improving coverage [14,58,59]. 

The question of “who and what is responsible for population patterns of health, disease, and wellbeing…” [35] 2/21/2020 3:49:00 PM (p. 694) is central to understanding the social determinants of health, and is a constant theme in the planning literature on sanitation [42,60,61]. For example, Hindu notions of purity and pollution have been well documented as a reason for persistent OD and a reluctance to handle one’s own fecal waste [32,62], and explain why people much prefer to build large pits. Nevertheless, waste management has been chronically underfunded under SBA-G [63], and the government’s recommended latrine design requires rural people to manage their own waste. The rural poor themselves had virtually no agency in designing or implementing sanitation policy. 

Our participants attributed this feeling of exclusion from the decision-making process, and the government’s decision not to invest in waste management, to a generalized disregard for rural areas. This is a perception rooted in the “social distance” that has been noted in the literature [36,62]. The failure to provide waste services was not considered accidental; it was spoken of as intentional, and one of many examples of how powerful agents design policies that result in the uneven distribution of public infrastructure [36,64]. Almost everyone we spoke to thought that public money went mainly to “big people” who lived in cities. That the government prioritizes the needs and interests of elites is not a new idea in India, and was, for example, a common critique of efforts aimed at making New Delhi a “green” and “clean” city [65,66,67]. Our participants reflected this perception, which in turn, led to generalized anger and distrust of many government initiatives, SBA-G included. This generalized lack of trust in government initiatives, and in the government at large, is well documented in the literature [67,68,69], but has not been discussed thus far as a part of the explanation for the slow uptake of SBA-G. 

Our study has some limitations that must moderate our conclusions. We worked with a small number of participants within one district in a low-income state. In a country as large as India, with cultural, social, economic, and political diversity, the results from our study do not allow us to make generalizable conclusions or policy recommendations. Nevertheless, as an exploratory study, the results from this study can be used to point to areas that require more research, be it quantitative or ethnographic, and to highlight assumptions in the current policy discourse that may not be warranted. For example, future studies with larger samples should explore how distrust of government initiatives overall inhibits uptake of latrines. This study also did not explore whether SBA-G has successfully spurred collective action around sanitation. Our observations and discussions would suggest that this is not happening, at least not in rural Bihar. However, this may be a sign of poor implementation or particular challenges due to socio-economic hierarchies in the state. Furthermore, our study did not include focus groups with women. Future research should explore the extent to which distrust of the government manifests among women, and how this might also affect latrine uptake by households. 

## 5. Conclusions

Based on interviews in rural Bihar conducted during India’s ambitious campaign to eliminate OD, our results emphasize the need for sanitation research that looks beyond household-level drivers to understand the social-structural determinants of latrine uptake. This could help researchers and policy makers to better understand the barriers to latrine uptake, even under an aggressive campaign such as Swachh Bharat Abhiyan. Our results undermine the common conflation of the observed persistence of OD with its alleged preference. We found that our participants knew that latrines are important for health and safety but were unable to build a latrine due to economic and structural factors. Our participants pointed out the ways in which the subsidy program needed redesigning. Our participants also expressed their distrust of the government; specifically, they talked about how the government prioritizes development in wealthier urban areas while ignoring the plight of “small” rural citizens, thus engendering feelings of distrust and resentment. Thus, we suggest there is a need for future studies, possibly in other states, that investigate these themes. Doing so could further elucidate the extent to which underlying social factors influence people’s ability to build household latrines and ensure that future policies are more effective. 

## Figures and Tables

**Table 1 ijerph-17-01384-t001:** Participant demographics.

Category	Description	Men	Women	Total
**Has household toilet**	Yes	2	2	4
	No	12	5	17
**Education**	None/Illiterate	2	4	6
	1st–8th grade	10	1	11
	9th–12th grade	1	1	2
	College and/or above	1	1	2
**Caste/Religion**	Scheduled Caste (SC)	5	3	8
	Other Backwards Caste (OBC)	8	1	9
	Gen	0	1	1
	Muslim	1	2	3
**Age**	18–24	1	4	5
	25–39	4	1	5
	40–65	4	1	5
	> 65	5	1	6
**Family size**	0–4 family members	5	2	7
	5–8 family members	7	4	11
	More than 8 family members	2	1	3

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
