# Peer review of "Understanding Open Defecation in the Age of *Swachh Bharat Abhiyan*: Agency, Accountability, and Anger in Rural Bihar"

_ijerph, 2020, doi:10.3390/ijerph17041384_

Round 1

Reviewer 1 Report

The manuscript is very informative and has important messages to be delivered. However, methods, results, and discussions are neither described nor presented soundly. Here are some suggestions:

1. The objectives of the paper are vague and should be defined clearly.

2. Provide more details on the criteria for questions and selecting interviewees.

3. The provision of questions is highly recommended. I invite the authors to provide the questionnaire as supplementary material.

4. More explanations are required about the "extended, semi-structured interviews." What are the outlines?

5. How many people participated in the survey? Seven men and six women?

6. The provision of comprehensive sociodemographic data of the participants is missing.

7. Results should be presented more scientifically. There is no sufficient statistical analysis. A sharper discussion requires sharper analysis.

8. Categorize and describe the barriers to use safe and improved sanitation systems. I can suggest technical, economic, and social barriers. Then you can specify how these barriers have contributions to sanitation challenges, which makes people defecate openly.

9. Again, only after conducting a sound statistical analysis, you may provide appropriate solutions in the conclusion section.

10. Describe the limitations of your research and provide suggestions for further studies.

Reviewer 2 Report

This manuscript provides a useful account of perceptions relating to open defecation and the SBA in Supaul District, Bihar. In general, more qualitative work is needed to understand sanitation in India, so this exploratory study is welcome. Your finding about the “perceived anti-poor and anti-rural bias,” which can lead to distrust of government sanitation programs and discourage participation stood out to me the most, and in my opinion, an in-depth qualitative exploration of this one theme would have made a strong paper in itself.

My overarching comment/concern is about the spectre of generalizability that is suggested throughout the paper. I caution the authors from generalizing about rural India in such broad strokes from such a small sample in what is correctly described as an “exploratory” study. Exploratory ethnographic studies are highly important and valuable, especially in WaSH, but this study still required a much larger sample size to broaden findings beyond such a small group. (1) A more focused interpretation of results, (2) more details about the sample and analytic procedure, and (3) a clear limitations section would strengthen the paper considerably. Detailed comments below all allude to these 3 overall comments.

1. Title: It is inappropriate to generalize to “rural India” in the title. At most, the title could say “…Supaul District, Bihar, India” (given a population of 2.2 million people, that is arguably still an extreme generalization, but it is the highest level I would consider to be appropriate).

Introduction:

2. Line 64-68: the way this is phrased suggests that these are contemporary thoughts on health education being necessary to improve sanitation, but the field has moved past that considerably in the past decade and definitely into more of a social marketing framework. I suggest adding some nuance about how researchers and practitioners USED TO think health ed mattered more than anything else.

2. Line 96: Suggest using a less judgmental word than “languished” when reporting a statistic.

3. Line 112-113: Again, I would caution from making it sound like researchers and practitioners still point to “lack of understanding” as the key factor—this is outdated. Many others in the past decade have studied and written about awareness being less of an issue than other factors (including behavioral and financial), hence the advent of CLTS and sanitation marketing as the primary approaches for rural sanitation today.

4. Line 123: I think the section “under the lid of SBA” should be moved earlier to Line 99 (before the “this paper builds on…” paragraphs.

5. Check repetition of citations, e.g. 35 and 50 are the same

Materials/Methods: More details need to be provided about the sample, as suggested below.

6. How many people were in each focus group?

7. How long did the interviews and focus group discussions last?

8. The rationale for conducting focus groups with men is explained clearly, but why was there not at least one focus group with women? If it was a logistical limitation, it should be stated as so. As I’m sure the authors know well, women in rural Bihar, while they may not feel like they have financial decision-making power, have a lot to say about sanitation and SBA that they may not otherwise share around men that would have added considerable nuance to this story.

9. Were any measures taken to consider power differentials in the focus groups or informal group meetings, especially when interviewing participants from different castes? In other words, were the focus groups same-caste groups or mixed? In my experience in rural Bihar, this can vastly affect what people are comfortable sharing in a group setting, particularly men. You say in the methods that you “ensured variation” in xyz factors, including caste. However, knowing what an important factor caste can be in rural Bihar, especially with regard to sanitation, more explanation is necessary to understand who these findings are coming from. In the discussion/limitations, there should be a discussion of how the background of participants may affect the transferability of the findings.

10. Line 190: how were sanitation and hygiene practices observed, and by whom? What role did this observation play in the analysis/findings?

11. Analysis: The paper needs at least a short paragraph (if not a sub-heading) on the qualitative analysis process. How were transcripts analyzed, and by whom? Was a coding framework used? As this was an exploratory study (one assumes with the aim of informing the authors’ future research), was any type of member checking conducted to bolster the veracity of interpretation?

Results

12. 3.1 on OD being a necessity: Did you ask participants about the presence of SBA-G (or TSC/NBA) activities that they had been exposed to in the past? Is there any possibility of their perceptions having changed as a result of behavior change/BCC activities? If you don’t believe this is the case, what makes you say so? The sample size is not sufficient to suggest that this is a more widespread issue in rural India as a whole; larger studies have documented the ‘preference’ piece quite convincingly, so rather than simply saying that your findings do not concur with them, it would be important to suggest reasons why you think this is the case. It would further the conversation on this very important factor.

13. 3.2 subsidies: Did you find evidence of any collective action-type activities that were used to address the subsidy/financial issue? If not, it appears that there was no clear collective mobilization effort that took place in these villages, which may be an important point to highlight. i.e. are people’s experiences here a result of poor implementation/no implementation, or something more structural?

14. 3.3 on trust: As mentioned earlier, this theme was the most interesting one to uncover. If at all possible, I would highly recommend that authors try to extract more quotes or examples from the existing data to flesh out this point in the paper. It is also obvious from the discussion section that the authors are well-versed in political ecology, and the sanitation field surely could benefit from deeper analysis through this lens. The observations on subsidies have been studied to death by others, but trust and power etc. requires political ecologists to join the field.

Discussion:

15. Your emphasis on structural determinants is important and much needed in sanitation. However, to truly be able to dig in ethnographically into these structural determinants, you would still need a much larger and varied sample size at different levels of a socio-ecological system to be able to triangulate your findings to some extent. This is an example of where the interpretation of results needs to remain within bounds. It’s difficult to make the case that 2 focus groups and interviews with 13 people in India of all places can uncover structural determinants that can be actionable beyond those 3 villages. For example, given the nature of your sample, it is fitting that a good portion of your results section still focuses on the role of knowledge, shame, etc. at the individual or community level, which doesn’t necessarily get at larger structural issues. I suggest that authors instead clearly draw the lines and make it clear that their findings suggest the need/help make a case for more rigorous/bigger ethnographic work that explores these structural determinants, rather than making the case that an exploratory study achieved that.

16. As any good qualitative paper will have, this paper needs a limitations section that clearly delineates the bounds/limits of what can be said from this sample, and suggest what future research could look like. Exploratory studies ideally lay out the groundwork for future research questions/objectives (and I would argue that the “trust” theme is a crucial and valuable theme to explore).

Reviewer 3 Report

Line 327-328, "hold" ?

Awesome work! Very interesting read and well-written! Thanks!

Round 2

Reviewer 1 Report

The authors addressed some of the comments and suggestions to some extent. Still, there is room for improvement.

Reviewer 2 Report

Thank you for addressing each of the concerns.